# An Ultra-Short Measure of Excessive Daytime Sleepiness Is Related to Circadian Biological Rhythms: The French Psychometric Validation of the Barcelona Sleepiness Index

**DOI:** 10.3390/jcm11133892

**Published:** 2022-07-04

**Authors:** Julien Coelho, Régis Lopez, Jacques Taillard, Emmanuel D’Incau, Guillaume Fond, Pierre Philip, Jean-Arthur Micoulaud-Franchi

**Affiliations:** 1Mixed Research Unit (UMR) 6033, Health Department, Medical Sciences, Sleep, Addiction, Neuropsychiatry (SANPSY), Bordeaux University, F-33000 Bordeaux, France; jacques.taillard@u-bordeaux.fr (J.T.); pierre.philip@u-bordeaux.fr (P.P.); jean-arthur.micoulaud-franchi@u-bordeaux.fr (J.-A.M.-F.); 2National Center for Scientific Research (CNRS), SANPSY, UMR 6033, F-33000 Bordeaux, France; edincau@free.fr; 3CHU Bordeaux, University Department of Sleep Medicine, CHU Bordeaux, F-33000 Bordeaux, France; 4Sleep Disorders Center, Department of Neurology, Gui-de-Chauliac Hospital, CHU Montpellier, F-34000 Montpellier, France; r-lopez@chu-montpellier.fr; 5INSERM U1061, Université Montpellier 1, F-34000 Montpellier, France; 6EA 3279: CEReSS—Centre d’Etude et de Recherche sur les Services de Santé et la Qualité de Vie, Marseille University, F-13000 Marseille, France; guillaume.fond@gmail.com; 7Hôpital La Conception, Assistance Publique des Hôpitaux de Marseille, F-13000 Marseille, France

**Keywords:** sleep, patient health questionnaire, sleepiness, circadian rhythm, psychometrics

## Abstract

The Barcelona Sleepiness Index (BSI) is an ultra-short instrument with several advantages for assessing excessive daytime sleepiness (EDS). The present study was performed to analyze the validity of the French version of the BSI in screening for EDS. We conducted a cross-sectional study on a population of students using an online questionnaire. The French version of the BSI was developed by a rigorous forward-backward translation process. We computed the discrimination properties of the BSI to predict EDS assessed by the Epworth Sleepiness Scale (ESS), as well as correlations with other sleep measures. In total, 662 students were enrolled in the study (mean age: 20.9 years, 76.0% women). The BSI score (mean = 1.5 ± 1.0) showed a strong correlation with the ESS score (*r* = 0.47, *p* < 10^−4^) and acceptable discrimination of EDS assessed by ESS score ≥ 11 (AUC = 0.742) with an optimal cutoff point of 2, as in the original study. The BSI score was significantly associated with sleep deprivation and social jetlag. Therefore, the French version of the BSI is a valid ultra-short instrument for EDS screening in individuals. In addition, the BSI score may be associated with both homeostatic and circadian processes. Further studies are needed to confirm these findings in general populations and in patients with sleep disorders.

## 1. Introduction

Excessive daytime sleepiness (EDS) is widespread in the general population [1], constitutes a common symptom of sleep disorders [2], and is associated with poor physical and mental health outcomes [3]. The severity of EDS can be objectively evaluated using direct electrophysiological recordings or indirect behavioral measures [4]; it can be subjectively evaluated using questionnaires [5,6,7]. Objective evaluations are suitable for clinical practice or research settings, while subjective evaluations are suitable for the assessment of sleepiness in epidemiological settings. The Epworth Sleepiness Scale (ESS), which consists of a questionnaire with questions related to the respondent’s propensity to fall asleep in eight specific situations scored on a scale of 0–3, is the most widely used and validated scale used for this purpose [5,8]. The questionnaire is easy to complete, differentiates among levels of EDS, and is sensitive to treatment-induced changes [9]. However, the design of the ESS has been criticized because some items are excessively general or almost identical; moreover, the total score focuses on sleep propensity, which is a subtype of EDS [10]. In addition, questionnaires that show stronger correlations with objective sleepiness tests would be useful [9], and the ESS does not consider variations in sleepiness throughout the day. Indeed, EDS reportedly has circadian rhythmicity based on subjective ratings [11] and electrophysiological recordings [12]; it should be systematically evaluated while considering the time of day. Circadian biological rhythms have demonstrated correlations with sleep and health outcomes; thus, they are important to consider in evaluations of sleep-wake disturbances [3]. The Time-Of-Day Sleepiness Scale (TODSS)—a long (24-item) scale that divides the ESS items and scores into the morning, afternoon, and evening categories—was created for this purpose [13]. Finally, despite its short length, the ESS may be excessively long for patients with sleep disturbances that can interfere with cognitive performance and attention [14]. The use of ultra-short questionnaires should be favored in such populations.

In recent decades, questionnaire design has improved with the establishment of focus groups during the design process [15]. The involvement of such groups allows the identification of real concerns of a specific group of people; it also considers the actual language and expressions used by these people, while reflecting a consensus among members of that group. The Barcelona Sleepiness Index (BSI) was developed and validated using this rigorous methodology in a three-step process [16]. Initially, 138 situations that could cause sleepiness were identified using focus groups. Then, 16 items were selected based on their psychometric properties. Finally, the BSI was developed from the two items that showed the greatest ability to predict objective sleepiness [16]. This screening instrument has several advantages. First, it is an ultra-short instrument that consists of only two items; such brevity and simplicity ensure fair and accessible use in the general population [17]. Second, it explores EDS while considering circadian biological rhythms through the examination of individuals in both the morning and afternoon [16]. Third, it shows stronger correlations with objective measures of sleepiness, compared with the ESS; it also has sufficient sensitivity to detect changes [16].

Considering the differences among populations in terms of medical conditions, as well as cultural and demographic factors (reflected in the variable cutoffs of questionnaires in different studies), the BSI should be validated for each language. Although French is the sixth most widely spoken language with 300 million speakers worldwide, the psychometric properties of the BSI have not been investigated in French-speaking adults. The present study was performed to analyze the validity of a French version of the BSI in screening for EDS after rigorous translation and validation with regard to sleep-wake timing, sleep disturbances, and mental health outcomes.

## 2. Materials and Methods

### 2.1. Setting and Participants

This cross-sectional study was conducted between December 2019 and January 2020 in a study population that consisted of medical students from Bordeaux University; such students represent a population at low risk of sleep breathing disorders but with high levels of EDS because of chronic sleep deprivation and circadian misalignment [18]. Various solicitation channels (e.g., email, management, and flyers) were used to maximize the response rate. The participants were informed of the research objective to evaluate sleep and asked to complete an Internet-based questionnaire.

### 2.2. Research Tools

The following sociodemographic data were recorded: age, sex, and year of study. The BSI is an ultra-short scale that evaluates EDS with scores of 0 (No), 1 (Yes, I feel sleepy BUT I do not fall asleep), 2 (Yes, I feel sleepy AND I fall asleep), or 3 (Yes, I fall asleep unexpectedly) for two items that explore different sleepiness-related situations (relaxing and standing inactive) at different times of day (morning and afternoon) [16]. The original version found that a cutoff score of ≥2 indicated severe EDS. The French version of the BSI was developed using a forward-backward translation method by two independent native French speakers and two independent native English speakers. We ensured the clarity and cultural acceptability of the French version of the BSI in a French student population by administering the instrument to 10 students. This preliminary test revealed no difficulties in understanding the items of the French BSI. No adaptations were required. The version of the French BSI used in this study is shown in Table 1. The score ranged from 0 to 6. Daytime disturbances assessed by the ESS and the Toronto Hospital Alertness Test (THAT) were used for convergent concurrent validation of the BSI [8,19]. The THAT is a 10-item self-reported questionnaire rated on a 6-point Likert scale, which is designed to measure perceived alertness during daytime; higher scores indicate greater alertness. A French version obtained after a rigorous translation process is available on sleepontario.com. Each scale referred to an EDS subtype: the ESS explored sleep propensity (similar to the BSI), while the THAT assessed alertness (the opposite of drowsiness). For divergent concurrent validation, insomnia symptoms were assessed with the Insomnia Severity Scale (ISI) [20], while anxiety and depressive symptoms were measured using the Patient Health Questionnaire 4 (PHQ-4) scale, a short 4-item instrument rated on a 3-point Likert scale with two items for each mental health symptom. A score ≥3 was considered indicative of anxiety or depressive symptoms [21]. Sleep-wake timing was studied for external validation of the BSI to investigate the impact of sleep deprivation and disrupted circadian biological rhythms on the BSI scores. The questions were based on the Munich ChronoType Questionnaire [22]. Participants were asked what time they usually went to bed (bedtime, “What time do you usually go to bed at night?”) and got up (rise time, “What time do you usually get up in the morning?”) and how many hours of actual sleep they achieved (sleep duration, “How many hours of actual sleep do you get at night?”) on workdays and on free days. These answers were then used to estimate their mean sleep efficiency (ratio of sleep duration over time-in-bed, defined as the difference between bedtime and rise time) [23]; their mean sleep duration (including workdays and free days), was regarded as a proxy for their homeostatic-dependent process; and their social jetlag (defined as the time difference between mid-sleep on workdays and mid-sleep on free days, with mid-sleep as the median between bedtime and rise time) [24], regarded as a proxy for their circadian-dependent process. Sleep deprivation was defined as <7 h per night on workdays, as recommended by the National Sleep Foundation [25]. Social jetlag with a shift of ≥2 h was considered significant [26].

### 2.3. Statistical Analysis

Data analysis was performed using R, version 4.1.2 (R Foundation for Statistical Computing, Vienna, Austria). Descriptive statistics were calculated as frequencies (%) for categorical variables and as means ± standard deviations for continuous variables. First, prediction by the BSI of EDS assessed by the ESS was evaluated by receiver operating characteristic analyses and areas under the curve (AUCs). Two ESS score cutoff points were considered (≥11 for significant EDS and ≥16 for severe EDS). We hypothesized acceptable discrimination (AUC 0.7–0.8), consistent with the original validation study [16]. Sensitivity, specificity, positive predictive value (PPV), negative predictive value (NPV), and Youden index (YI) were computed. The cutoff point that maximized the YI was regarded as the best performance. We expected a cutoff point of 2, consistent with the original study [16]. Then, we estimated correlations of the BSI score with sleep and mental health outcomes, using Pearson’s correlation coefficient with a significance level of 5%. To control for the increase in type I error associated with multiple comparisons, all *p*-values were corrected by the Benjamini–Hochberg method [27]. We estimated correlations of the BSI score with daytime disturbances assessed by the ESS and the THAT. We hypothesized medium (>0.4) to high (>0.6) degrees of correlation, consistent with the original convergent concurrent validation study [16,28]. We estimated correlations of the BSI score with insomnia, anxiety, and depressive symptoms (ISI and PHQ-4). We hypothesized low (>0.2) to medium (>0.4) degrees of correlation, consistent with previous divergent concurrent ESS validations [29,30]. For the BSI external validation process, we specifically analyzed the correlations of the global BSI score and the two separate BSI items (morning and afternoon) with sleep-wake timing, specifically mean sleep duration (hh:mm) and social jetlag (hh:mm). For comparison, correlations with ESS and THAT scores were also calculated.

## 3. Results

### 3.1. Sample Description

The study population consisted of 662 students, representing 13.8% of the theoretical enrollment of 4807 students in the first and second cycles of medical studies at Bordeaux University. The mean age of the participants was 20.9 years (standard deviation = 2.6 years), 76.0% were women, and 41.4% were freshmen (first year of study). Typical sleep-wake timings were 23:30 to 07:34 on workdays with 7 h 25 min of sleep and 23:57 to 09:07 on free days with 8 h 20 min of sleep (Appendix A). In total, 180 students (27.2%) were classified as sleep-deprived (<7 h) and 68 (10.3%) had significant social jetlag (≥2 h). Additional information concerning sleep-wake timing, sleep disturbances, and mental health outcomes are presented in Table 2. The BSI score ranged from 0 to 6 with a mean of 1.5 (±1.0); 309 students (46.7%) had a score ≥2 (Figure 1).

### 3.2. Ability of BSI Score to Predict EDS

Prediction of EDS according to ESS score was better when using a cutoff point of 11 (AUC = 0.742) than when using a cutoff point of 16 (AUC = 0.681); discrimination was considered acceptable overall (Appendix A). The BSI cutoff point of 2 maximized the YI at both ESS cutoff points with high PPV (82%) for discriminating EDS (ESS score ≥ 11) from non-EDS (ESS score < 11) and high NPV (82%) for discriminating severe EDS (ESS score ≥ 16) from non-severe EDS (ESS score < 16) (Table 3).

### 3.3. Concurrent and External Validity of the BSI

BSI concurrent and external validity results are presented in Table 4. With regard to convergent validation, the correlations of BSI score with ESS and THAT scores were medium and highly significant (*r* = 0.47, *p* < 10^−4^ and *r* = −0.40, *p* < 10^−4^, respectively). With regard to divergent validation, the correlations of the BSI score with insomnia, anxiety, and depressive symptoms were low but significant (*r* = 0.36, *p* < 10^−4^; *r* = 0.15, *p* < 10^−4^; *r* = 0.24, *p* < 10^−4^). With regard to external validity, the correlations of the BSI score with sleep-wake timing were all significant. The afternoon item showed weak correlations with mean sleep duration (*r* = 0.11, *p* = 0.006) and mean sleep efficiency (*r* = −0.09, *p* = 0.017), while the morning item showed no correlations. Conversely, the afternoon item showed no correlation with social jetlag, while the morning item showed a weak correlation (*r* = 0.13, *p* = 0.001). Indeed, social jetlag of ≥ 2 h was present in 18 of the 111 students (16.2%) who would fall asleep when relaxing in the morning vs. 50 of the 551 students (9.1%) who would not fall asleep (*p* = 0.024). Note that the correlation with sleep-wake timing was greater for the BSI score than for the ESS score, while the correlations with mean sleep duration and mean sleep efficiency were lower for the BSI score than for the THAT score.

## 4. Discussion

In this sample of medical students with high rates of chronic sleep deprivation and circadian misalignment, we found that the French version of the BSI showed good psychometric properties in terms of cutoff, as well as convergent and divergent concurrent validation. The BSI showed the same cutoff (score ≥ 2) as the original study for severe EDS. The BSI score showed stronger correlations with closed constructs (daytime sleepiness of the ESS and alertness during daytime) than with more distant constructs (insomnia, anxiety, and depressive symptoms). The score of the French version of the BSI was strongly correlated with the score of the ESS (*r* = 0.47, *p* < 0.001), the most widely used validated scale in clinical and research settings. This result was consistent with the original study (*r* = 0.52, *p* < 0.001) conducted in an older population of patients with sleep breathing disorders [16].

Notably, in contrast to the ESS, this study showed that the BSI score exhibited weak but significant correlations with sleep-wake timing. When BSI components were considered separately, the morning item was significantly associated with social jetlag, a proxy for the circadian-dependent process, while the afternoon item was significantly associated with mean sleep duration, a proxy for the homeostatic-dependent process. Therefore, this study showed that the score in an ultra-short instrument for EDS may be associated with both homeostatic and circadian processes.

In particular, social jetlag, a chronic disturbance of the circadian system observed frequently among individuals with evening chronotype [31], was associated with a twofold increased risk of morning EDS. This result was consistent with the synchrony effect framework that describes different circadian rhythmicity of EDS (superior performance at optimal times of day and inferior performance at suboptimal times of day) depending on an individual’s chronotype [32]. It is important to note that the sleepiness-related situation also differed between the two items of the BSI: lower arousal for the morning item and higher arousal for the afternoon item. These differences presumably influenced the assessment of EDS, similar to the findings with objective measures [33]. However, the circadian rhythmicity of EDS does not appear to depend on the type of measurement (i.e., objective or subjective) [34].

These considerations highlight the importance of considering circadian rhythmicity when measuring EDS; they support the use of a sleepiness scale that considers biological rhythms. Thus far, only the TODSS has addressed this issue [13]. This scale divides the ESS items and scores into the morning, afternoon, and evening categories. However, it has several limitations. First, the TODSS is a long (24-item) scale, which may call on the abstraction abilities of the respondents several times. Indeed, they must project themselves into activities at different times of the day, with a reduced probability of prior exposure to such situations. Second, impacts on biological rhythms (e.g., sleep-wake timing, sleep complaints, and mental health outcomes) were not estimated. Third, there has been no validation of the TODSS in the general population. The BSI constitutes a suitable ultra-short instrument for the identification of EDS in a population of young adults at risk of sleep deprivation or disrupted biological rhythms. Further studies are needed to confirm these findings in a general population with individuals of different ages and socio-professional categories (with consideration of sex) and in patients with other sleep disorders, including sleep breathing disorders as in the original validation study [16].

Indeed, the BSI score exhibited weak but significant correlations with mean sleep efficiency, a measure of nocturnal awakening that is frequently altered in sleep disorders (e.g., sleep breathing disorders, chronic insomnia, periodic sleep movements) [35,36]. It would be useful to evaluate the associations of the BSI score and each of the scale items among patients with different sleep disorders, including patients with rare hypersomnia’s that combine different subtypes of EDS (e.g., drowsiness, sleep inertia, sleep attacks, and long sleep time). In addition, the consideration of sleep disturbances overall, such as by combining the ESS with a global tool (e.g., the Pittsburgh Sleep Quality Index [37]), could provide a better understanding of interactions between the BSI score and sleep disturbances.

This study had some limitations. First, medical students were included on a voluntary basis, which limited the representativeness of our sample. Women were overrepresented (76.0% vs. 65.0% in French medical studies), while freshmen were underrepresented (41.4% vs. 65.1% among all medical students at Bordeaux University). However, additional analyses showed consistent results across these different subgroups (Appendix A). Moreover, our sample showed a 27.2% prevalence of chronic sleep deprivation, which was similar to the prevalence of short sleep duration (<7 h) in studies conducted among medical students in different countries (from 24% in the United Kingdom to 49% in Taiwan) [38]. Similarly, the prevalence of significant social jetlag (≥2 h) were equivalent between our sample (10.3%) and a 2020 Dutch general population survey (8%) [39]; the prevalence of insomnia in our study (22.0%) was equivalent to the prevalence in a 2020 representative study performed in a general population (21%) [40]. Nevertheless, our study participants were at high risk of severe EDS (28.1%) and the results cannot be extrapolated to the generation population. Despite this limitation, medical students are particularly affected by sleep hygiene problems that can induce EDS, and they constitute a suitable population for preventive actions on sleep behavior that could benefit from the use of this scale [41]. Second, there were no subjective assessments of sleepiness at the wheel or objective measures of EDS for external validity. Therefore, further studies are needed to evaluate the predictive value of the BSI with electrophysiological recordings, as in the original study [16], along with the risk of motor vehicle accidents related to sleepiness, which is the main direct consequence of EDS [42]. Such studies are particularly important because none of the BSI items explore accidental risk situations, as in the ESS. The relationships between the BSI score and each of the scale items with overall assessments of function would also be useful. Third, there was no evaluation of medical and psychiatric status, legal and illegal consumption, and comorbidities associated with sleep disorders (e.g., sleep breathing disorders) to consider the effects of such factors in external validation of the French version of the BSI. Although our sample comprised a population at low risk of sleep disorders, substance use disorders and mental health complaints are common in this population and should be considered when assessing sleep, EDS, and circadian biological rhythms [43,44]. Fourth, test-retest repeatability and sensitivity to change were not studied. However, the BSI represents a favorable tool for measuring and monitoring sleepiness in large samples, in an epidemiological context, or as part of a sleep hygiene intervention. Further studies are needed to explore the sensitivity of the BSI score to change in the general population.

## 5. Conclusions

The French version of the BSI scale, produced by rigorous translation and validation processes regarding sleep-wake timing, sleep complaints, and mental health outcomes, is a valid ultra-short instrument for EDS screening. These results highlight the importance of considering homeostatic pressure, circadian rhythmicity, and sleepiness-related situations when measuring EDS.

## Figures and Tables

**Figure 1 jcm-11-03892-f001:**
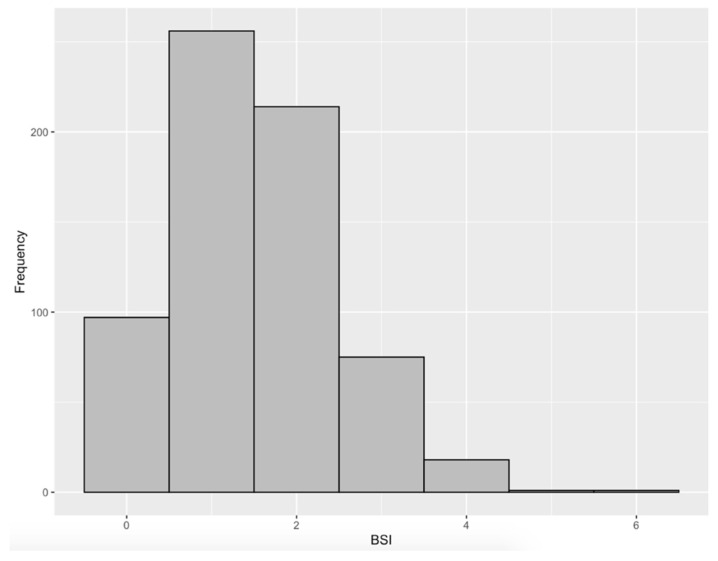
Frequency histogram of the BSI score (*n* = 662).

**Table 1 jcm-11-03892-t001:** French translation and description of the Barcelona Sleepiness Index (*n* = 662).

These questions refer to the sleepiness experienced during recent weeks		No	Yes, I feel sleepy BUT I do not fall asleep	Yes, I feel sleepy AND I fall asleep	Yes, I fall asleep unexpectedly
Ces questions font référence à la somnolence ressentie dans les semaines précédentes		Non	Oui, Je me sens somnolent·e MAIS je ne m’endors Pas	Oui, je me sens Somnolent·e ET je m’endors	Oui, je m’endors subitement
In the morning, when relaxing	Le matin, en se reposant	228 (34.4%)	323 (48.8%)	102 (15.4%)	9 (1.4%)
In the afternoon, when standing, inactive in a public place (waiting to be served, or for the bus, subway, or a friend, etc.)	Dans l’après-midi, en se tenant debout, inactif·ve dans un lieu public (en faisant la queue, ou en attendant le bus, le métro, ou un ami, etc.)	266 (40.2%)	356 (53.8%)	38 (5.7%)	2 (0.3%)

**Table 2 jcm-11-03892-t002:** Descriptive characteristics (*n* = 662).

	Total *n* = 662	Mean ± Standard Deviation (Min:Max)
Age (years)		20.9 ± 2.6 (16:35)
Sex		
− Male	159 (24.0%)	
− Female	503 (76.0%)	
Year of study		
− Freshmen	274 (41.4%)	
− Others	388 (58.6%)	
Bedtime on workdays (hh:mm)		23:30 ± 51 min
Rise time on workdays (hh:mm)		07:34 ± 45 min
Sleep duration on workdays (hh:mm)		7 h 25 ± 54 min
− 7 h	482 (72.8%)	
− 6–7 h	144 (21.8%)	
− <6 h	36 (5.4%)	
Bedtime on free days (hh:mm)		23:57 ± 68 min
Rise time on free days (hh:mm)		09:07 ± 76 min
Sleep duration on free days (hh:mm)		8 h 27 ± 100 min
Mean sleep duration (hh:mm)		7 h 43 ± 53 min
Mean sleep efficiency (mean percentage)		92% ± 3% (73%:97%)
− <85%	17 (2.6%)	
− 85–95%	566 (85.5%)	
− ≥95%	79 (11.9%)	
Social jetlag (hh:mm)		
− Significant (≥2 h)	68 (10.3%)	55 ± 47 min (−85:225)
Insomnia Severity Scale (ISI)		
− Score ≥ 15	146 (22.0%)	10.1 ± 5.3 (0:28)
Epworth Sleepiness Scale (ESS)		12.5 ± 5.2 (0:24)
− Score ≥ 11	423 (63.9%)	
− Score ≥ 16	186 (28.1%)	
Anxiety symptoms (PHQ-4)		2.5 ± 1.9 (0:6)
− Score ≥ 3	277 (41.8%)	
Depressive symptoms		2.0 ± 1.7 (0:6)
− Score ≥ 3	211 (31.9%)	
Barcelona Sleepiness Index (BSI)		1.5 ± 1.0 (0:6)
− Score ≥ 2	309 (46.7%)	

**Table 3 jcm-11-03892-t003:** Ability of BSI score to predict EDS, according to ESS score.

	BSI Score	Prevalence	Sensitivity	Specificity	PPV	NPV	YI
ESS score ≥ 11	≥1	85%	94%	29%	70%	72%	0.23
	≥2	47%	60%	77%	82%	52%	0.37
	≥3	14%	22%	99%	98%	42%	0.21
ESS score ≥ 16	≥1	85%	96%	19%	32%	92%	0.15
	≥2	47%	66%	61%	40%	82%	0.27
	≥3	14%	28%	91%	56%	77%	0.19

BSI: Barcelona Sleepiness Index; EDS: Excessive Daytime Sleepiness; ESS: Epworth Sleepiness Scale; PPV: positive predictive value; NPV: negative predictive value; YI: Youden index.

**Table 4 jcm-11-03892-t004:** Correlations of BSI score (and each item on the scale) with sleep outcomes (*n* = 662).

	BSI Score	ESS Score	THAT Score	ISI Score	Anxiety Symptoms	Depressive Symptoms	Mean Sleep Duration	Mean Sleep Efficiency	Social Jetlag
BSI Morning item	0.80 *	0.35 *	−0.31 *	0.30 *	0.10 ^#^	0.16 *	−0.07	−0.07	0.13 *
BSI Afternoon item	0.69 *	0.35 *	−0.29 *	0.22 *	0.13 *	0.20 *	−0.11 ^#^	−0.09 ^† ^	−0.02
BSI score	-	0.47 *	−0.40 *	0.36 *	0.15 *	0.24 *	−0.11 ^#^	−0.10 ^#^	0.09 ^† ^
ESS score	-	-	−0.34 *	0.27 *	0.17 *	0.21 *	−0.08	−0.05	0.01
THAT score	-	-	-	−0.56 *	−0.46 *	−0.53 *	0.17 *	0.22 *	0.03

ESS: Epworth Sleepiness Scale; THAT: Toronto Hospital Alertness Test; ISI: Insomnia Severity Index Pearson’s correlation coefficient: ^†^
*p* < 0.05, ^#^
*p* < 0.01, * *p* < 0.001.

## Data Availability

Not applicable.

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
