# Peer review of "An Ultra-Short Measure of Excessive Daytime Sleepiness Is Related to Circadian Biological Rhythms: The French Psychometric Validation of the Barcelona Sleepiness Index"

_jcm, 2022, doi:10.3390/jcm11133892_

Round 1

Reviewer 1 Report

The study by Coelho et al. addresses the validation of the Barcelona Sleepiness Index (BSI) in a cohort of students. The study is very well designed and includes a large and largely representative if restricted cohort (general population, young age). The authors find good performance of the BSI in this cohort and state that the BSI is a potential primarily brief tool for assessing daytime sleepiness that can provide information on discriminative strength not only in SDB but to circadian disruption and sleep deprivation. They show that the BSI has additional discriminative properties to distinguish  sleep deprivation in addition to the ESS. They also mention the important aspect of time of day, which is not captured by the ESS, and interpret it in context. 

The paper is well written and understandable. In the presentation of the results, it might be useful to really limit to the highlights, as the number of figures tables is quite high. 

The authors recognize the limitations of the study and formulate them comprehensively. 

I think the work is very successful. For me, Figure 1 is not absolutely necessary, as the authors already address it in the text and could also add this information in the text. Or also Figure and Table 3, which could be summarised here so that the amount of figures and tables is somewhat reduced and thus more focused.

Author Response

The study by Coelho et al. addresses the validation of the Barcelona Sleepiness Index (BSI) in a cohort of students. The study is very well designed and includes a large and largely representative if restricted cohort (general population, young age). The authors find good performance of the BSI in this cohort and state that the BSI is a potential primarily brief tool for assessing daytime sleepiness that can provide information on discriminative strength not only in SDB but to circadian disruption and sleep deprivation. They show that the BSI has additional discriminative properties to distinguish sleep deprivation in addition to the ESS. They also mention the important aspect of time of day, which is not captured by the ESS, and interpret it in context. The paper is well written and understandable. The authors recognize the limitations of the study and formulate them comprehensively. I think the work is very successful.

Authors: We thank the reviewer for their encouraging words, and for providing us with helpful comments.

In the presentation of the results, it might be useful to really limit to the highlights, as the number of figures tables is quite high. For me, Figure 1 is not absolutely necessary, as the authors already address it in the text and could also add this information in the text. Or also Figure and Table 3, which could be summarised here so that the amount of figures and tables is somewhat reduced and thus more focused.

Authors: We thank the reviewer for pointing out this lack of readability in our results. We have refocused on the main findings. Some elements presented in the tables have been removed from the text. Two figures (Figures 1 and 3) have been relegated to Supplementary Materials.

Pages 4-5, Line 167-172: “Typical sleep–wake timings were 23:30 to 07:34 on workdays with 7 h 25 min of sleep and 23:57 to 09:07 on free days with 8 h 20 min of sleep (Supplementary Figure S1). In total, 180 students (27.2%) were classified as sleep-deprived (< 7 h) and 68 (10.3%) had significant social jetlag (≥ 2 h). Additional information concerning sleep–wake timing, sleep dis-turbances, and mental health outcomes is presented in Table 2.”

Pages 6, Line 181-184: “The BSI cutoff point of 2 maximized the YI at both ESS cutoff points with high PPV (82%) for discriminating EDS (ESS score ≥ 11) from non-EDS (ESS score < 11) and high NPV (82%) for discriminating severe EDS (ESS score ≥ 16) from non-severe EDS (ESS score < 16) (Table 3).”

Reviewer 2 Report

This is an interesting manuscript investigating Excessive Daytime Sleepiness using Barcelona Sleepiness Index in French medical students. However, even though this is an interesting approach, there are several issues that have to be addressed

Authors should calculate the response rate or add a preliminary sample size analysis.

Explain the procedures used to ensure that the imbalance of the sample in terms of gender and freshman does not bias the results. Is there anything that could justify the bias towards female respondents in this study?

Table 2 - are in the group of Sleep deprivation less than 7 hours calculated those students belonging to the group of Sleep deprivation less than 6 hours? That needs some clarification.

Lines 224-226 - appropriate references should adequately support statements.

Would you expect a similar correlation with BSI using the Pittsburgh Sleep Quality Index instead of the Epworth sleepiness scale?

The writing style has to be revised. A native English speaker could assist the writers on that.

Author Response

This is an interesting manuscript investigating Excessive Daytime Sleepiness using Barcelona Sleepiness Index in French medical students. However, even though this is an interesting approach, there are several issues that have to be addressed.

Authors: We thank the reviewer for pointing out these issues.

Authors should calculate the response rate or add a preliminary sample size analysis.

Authors: Unfortunately, we were unable to retrieve the number of students who were contacted (email distributed, opened, or flyer distributed). However, the response rate can be approximated using the theoretical number of students per year (eligible population[1]). This rate remains imprecise because the number of students per year may vary according to repetitions or moves, and because recruitment probably did not include all of the students. Nevertheless, the calculations indicate an overall response rate of 13.8%, with a better representation from upper-year students (Table 1). We have added these considerations to the Results section.

Table 1. Response rate by year of study

All

1st year

2nd year

3rd year

4th year

5th year

6th year

Included

662

274

50

83

65

99

91

Eligible

4807

3131

340

334

334

334

334

Response rate

13.8%

8.8%

14.7%

24.9%

19.5%

29.6%

27.2%

Page 4, Lines 164-166: “representing 13.8% of the theoretical enrollment of 4807 students in the first and second cycles of medical studies at Bordeaux University.”

Explain the procedures used to ensure that the imbalance of the sample in terms of gender and freshman does not bias the results. Is there anything that could justify the bias towards female respondents in this study?

Authors: As shown above, freshmen far outnumbered students in other years. Thus, they were “underrepresented” rather than “overrepresented” in our sample (41.4% observed vs. 65.1% theoretical). Women represent 65.0% of enrollment in medical schools according to a 2015 survey[2]. Therefore, they were rather overrepresented in our sample, but this is a classic overrepresentation in studies based on voluntary work[3]. Moreover, as explained in the Discussion, our sample had a similar prevalence of chronic sleep deprivation, significant social jetlag, and insomnia to that of previous studies on similar populations, suggesting that our sample was not biased for the variables of interest.

We also performed subgroup analyses to ensure that the gender and freshman imbalances in the sample did not bias the results. These analyses showed high consistency in the associations between the BSI and the ESS, the THAT, the ISI, and depressive symptoms. The associations between mean sleep duration, mean sleep efficiency, and social jetlag were not always significant for anxiety symptoms, although the estimates remained close to the overall values (except for anxiety symptoms and social jetlag in males, where the associations disappeared). We have added text related to these considerations to the Discussion of our manuscript.

Table 2. Subgroups analyses

All

(n = 662)

Female (n = 503)

Male

(n = 159)

Freshmen

(n = 274)

Others

(n = 388)

ESS

  0.47*

 0.49*

 0.41*

 0.44*

 0.49*

THAT

−0.40*

−0.42*

−0.33*

−0.42*

−0.39*

ISI

  0.36*

 0.41*

 0.17°

 0.37*

 0.35*

Anxiety symptoms

  0.15*

 0.20*

       −0.04

 0.17#

 0.14#

Depressive symptoms

  0.24*

 0.25*

 0.22#

 0.24*

 0.24*

Mean sleep duration

−0.11#

−0.13#

−0.07

−0.13°

−0.10°

Mean sleep efficiency

−0.10#

−0.17*

 0.08

−0.16#

−0.07

Social jetlag

  0.09°

 0.12#

 0.01

 0.11

 0.08

p < 0.05°

p < 0.01#

p < 0.001*

Pages 7-8, Lines 260-264: “Women were overrepresented (76.0% vs. 65.0% in French medical studies), while freshmen were underrepresented (41.4% vs. 65.1% among all medical students at Bor-deaux University). However, additional analyses showed consistent results across these different subgroups (Supplementary Table S1). Moreover,”

Table 2 - are in the group of Sleep deprivation less than 7 hours calculated those students belonging to the group of Sleep deprivation less than 6 hours? That needs some clarification.

Authors: It was all of the subjects sleeping less than 7 hours and not just those sleeping between 6 and 7 hours. We have changed the text. Thank you.

Page 5, Line 174:

Table 2. Descriptive characteristics (n=662).

n (%)

Mean ± SD {min;max}

Sleep duration on workdays (hh:mm)

-                 > 7h

-                 6-7h

-                 < 6h

662 (100%)

482 (72.8%)

144 (21.8%)

36 (5.4%)

7 h 25 ± 54 min

Lines 224-226 - appropriate references should adequately support statements.

Authors: We thank the reviewer for this comment, which allowed us to add an important reference to explain this sentence: Sleep timing, chronotype and social jetlag: Impact on cognitive abilities and psychiatric disorders – Biochemical Pharmacology - 2021. As explained in this review: “Social jetlag is greater in evening types and confirmed that late chronotypes showed the largest differences in sleep timing between work and free days, thus creating a considerable sleep debt on work days for which they compensated on free days. In late chronotypes, this social jetlag is essentially linked to their phase-delayed biological clock. In late chronotypes, the circadian sleep window occurs late after the onset of melatonin secretion”. This reference has been added to the corresponding sentence.

Page 8, Line 223-225: “In particular, social jetlag, a chronic disturbance of the circadian system observed frequently among individuals with evening chronotype [31], was associated with a twofold increased risk of morning EDS.”

Would you expect a similar correlation with BSI using the Pittsburgh Sleep Quality Index instead of the Epworth sleepiness scale?

Authors: The PSQI is a scale for measuring sleep disturbances. It uses 7 subscales that add up to a single overall score. Subscale number 7 concerns poor form during the day based on two questions: one about daytime sleepiness and the second about daytime fatigue. We expected that the BSI would be highly correlated with these two questions and this subscale (> 0.4). The correlation on the other subscales (sleep latency, sleep efficiency, sleep duration, and sleep quality) and on the overall score was expected to be significant but weaker (< 0.4), as found in this study. Although an analysis of the correlation between the BSI and PSQI would be interesting for research purposes, the Epworth Sleepiness Scale is a much more frequently used tool in the clinical practice of sleep medicine. This is why the Epworth Sleepiness Scale was chosen as the reference in this study. We have added some text to the Discussion to put the comparison with the PSQI into perspective.

Page 8, Line 255-258: “In addition, the consideration of sleep disturbances overall, such as by combining the ESS with a global tool (e.g., the Pittsburgh Sleep Quality Index [37]), could provide a better understanding of interactions between the BSI score and sleep disturbances.”

The writing style has to be revised. A native English speaker could assist the writers on that.

Authors: The article has undergone complete English editing. Many writing style elements have been corrected throughout the manuscript. Particular attention has been given to ensure that the original meaning was retained. certificates are available at the links below.

http://www.textcheck.com/certificate/iB3FGi

http://www.textcheck.com/certificate/y4qq1x

Other fixes:

Page 2, Line 84: “This cross-sectional study was conducted between December 2019 and January 2020”

[1]https://www.medshake.net/PACES/numerus-clausus/bordeaux/

[3]https://www.nature.com/articles/s41598-017-02232-y
